# The REGENERATE Study: A Non-Randomized Feasibility Study of an Intervention to REduce anticholinerGic burdEN in oldER pATiEnts

**DOI:** 10.3390/geriatrics9060152

**Published:** 2024-11-25

**Authors:** Athagran Nakham, Christine Bond, Moira Cruickshank, Rumana Newlands, Phyo Kyaw Myint

**Affiliations:** 1Faculty of Pharmaceutical Sciences, Neresuan University, Phitsanulok 65000, Thailand; 2Ageing Clinical and Experimental Research (ACER) Team, Institute of Applied Health Sciences, School of Medicine, Medical Sciences and Nutrition, University of Aberdeen, Aberdeen AB25 2ZD, UK; phyo.myint@abdn.ac.uk; 3School of Medicine, Medical Sciences and Nutrition, University of Aberdeen, Aberdeen AB25 2ZD, UK; c.m.bond@abdn.ac.uk; 4Aberdeen Centre for Evaluation, School of Medicine, Medical Sciences and Nutrition, University of Aberdeen, Aberdeen AB25 2ZD, UK; m.cruickshank@abdn.ac.uk (M.C.); r.newlands@abdn.ac.uk (R.N.)

**Keywords:** anticholinergic burden, deprescribing, feasibility study

## Abstract

**Background:** Anticholinergic burden (ACB) from medications has been associated with adverse outcomes in older adults. **Aim:** The aim was to conduct a non-randomized feasibility study of an intervention to reduce the anticholinergic burden in older patients (REGENERATE) to inform a subsequent definitive trial. **Methods:** The development and evaluation of an ACB reduction intervention was guided by the Medical Research Council framework. Findings from preliminary studies, two systematic reviews, and two qualitative studies informed the design of a mixed-method feasibility study. The study was conducted in one UK primary care site. The clinical pharmacist identified and invited potentially eligible patients, reviewed their medications, and made recommendations to reduce the ACB as needed. Patients completed surveys at baseline and 6 and 12 weeks post-intervention. A purposive sample of patients and healthcare professionals was interviewed. **Results:** There was a response of 16/20; 14/16 attended the pharmacist-led consultation and completed the baseline questionnaire, and 13/14 completed both follow-up questionnaires. The sustainability of deprescribing was confirmed. The results suggest the potential of the intervention to reduce side effects from medications and improve quality of life (EQ-5D-5L). The interviews showed patients were happy with the study processes and the medication changes and were satisfied with the pharmacist’s consultation. **Conclusions:** This feasibility study demonstrated that a deprescribing/reducing ACB intervention in older adults is feasible in a primary care setting and may benefit patients. Well-designed RCTs and cost-effectiveness studies should be undertaken to confirm the benefits of ACB deprescribing in primary care settings.

## 1. Introduction

The population is aging, which is associated with a gradual decrease in physical and mental health coupled with a growing risk of disease and multi-morbidity [1]. Hence, polypharmacy in older adults is common, leading to more side effects from medicines and drug interactions [2]. Medications with anticholinergic properties are commonly prescribed in older adults for a wide variety of common clinical conditions including Parkinson’s disease, depression, chronic obstructive pulmonary disease, urinary incontinence, and allergic rhinitis [3,4,5]. Their use is steadily increasing with estimates varying from 37% to 63% of the population depending on the clinical setting [6,7,8].

Anticholinergic medicines, as their name implies, block acetylcholine neurotransmission in the brain and peripheral organs resulting in effects related to their therapeutic purpose, as well as unwanted adverse effects, some of which may be longer term and less visible [9]. The term “anticholinergic burden” or “ACB” refers to the cumulative anticholinergic action resulting from the concomitant use of one or more anticholinergic medicines. A growing body of evidence suggests that a high anticholinergic burden (ACB) is linked to adverse outcomes such as poor physical functioning, dementia, falls, and mortality in older adults [10]. There is limited published research on the clinical effectiveness of interventions to reduce ACB [11].

There has recently been an interest in ensuring all medicine use is appropriate and regularly reviewed. Deprescribing is the process of medication withdrawal or dose reduction to correct or prevent medication-related complications, improve outcomes, and reduce costs [12]. Both general deprescribing and ACB drug deprescribing have the goal of reducing medication burden, risk of falls, hospitalization, and death and improving and/or preserving cognitive function [13,14]. Deprescribing (i.e., removing medicines no longer needed) in line with current best practice is now well established in the UK and supported by robust research evidence. However, few, if any, definitive trials of ACB reduction deprescribing initiatives have been published [15,16,17,18,19]. Long-term prescribing of high-scoring ACB drugs should be reviewed, and where possible, the drugs should be deprescribed. If the therapeutic need remains, alternative treatments should be explored, and if none exist, a reduced dosage should be considered.

To address this gap in research, a feasibility study was developed and evaluated following the Medical Research Council framework for complex interventions [20]. The preliminary work involved a quantitative systematic review [11], a qualitative systematic review [21], and qualitative interviews with patients and health care professionals [22]. The findings of these preliminary studies were combined into a logic model (Appendix A), which informed the design of a mixed-method feasibility study to investigate remaining areas of uncertainty that needed to be resolved prior to conducting a definitive randomized controlled trial. These areas of uncertainty included the acceptability and ability of an independent pharmacist prescriber to undertake the medication review, potentially deprescribing the ACB drug, and the sustainability of the changed regimen. Areas of research uncertainty, such as the willingness of patients to be recruited and participate in the study processes, were also explored.

Thus, our aim was to conduct a non-randomized feasibility study of an intervention to REduce anticholinerGic burdEN in oldER pATiEnts (REGENERATE) to inform a subsequent definitve trial.

## 2. Materials and Methods

This single-arm, non-randomized feasibility study was conducted in one primary care setting (GP practice) in the Grampian Health Board area, Scotland, UK, with a registered population of 8231 people. The study was conducted from July 2021 to February 2022. The trial was registered on ClinicalTrials.gov, no. NCT04660838, and was approved by the North of Scotland Research Ethics committee and Grampian Research and Development.

The objectives were to test processes for patient participant identification, assess the recruitment and retention rates of patient participants, review potential outcome measures and data collection processes, and explore the intervention’s acceptability to patients and healthcare professionals.

### 2.1. Participant Inclusion Criteria, Identification, and Recruitment

Patients were eligible for inclusion if they were ≥65 years of age, on one or more long-term drugs (prescribed for a minimum of 6 weeks) with ACB potential (ACoB scale ≥ 3; defined according to Boustani et al. [23]), and able to provide informed consent. We chose the anticholinergic cognitive burden (AcoB) scale as it has been commonly used in routine practice and is well documented. Furthermore, it includes relevant thresholds of ACB score linked to adverse outcomes. For example, a score ≥ 3 is associated with a 46% decline in cognitive ability over 6 years, and each one-point increase in the total score is correlated with a 26% increased risk of death. The exclusion criteria were patients with severe mental illness (such as the diagnosis of severe anxiety, severe depression, severe dementia, etc.), patients who were terminally ill (life expectancy less than 6 months), or patients who in the opinion of a responsible clinician were not suitable for participation or were taking part in another similar study.

The study site was identified through personal networks. The pharmacist initially applied a computer search to a sample of 200 patients, randomly selected from the practice records, to identify those ≥65 years of age and having an anticholinergic cognitive burden scale ≥ 3. A pharmacist and GP screened the list of those potentially eligible against the full inclusion and exclusion criteria and maintained a log of exclusions with reasons. Initially, a maximum variation sample of 20 patients were invited from the screened list to include a range of conditions, sexes, and ages. It was estimated this would be sufficient to meet target recruitment numbers. Other patients would be approached as necessary. Patients were first contacted by phone by the pharmacist to explain the study and then mailed an invitation pack from the GP practice (signed by the delegated primary care pharmacist). A stamped addressed envelope was included for returning an expression of interest form to the research team. The main researcher, A.N., phoned the patients to give them an opportunity to ask questions and took their verbal recorded informed consent. The names and IDs (assigned for each individual patient) of those agreeing to take part were forwarded to the primary care pharmacist (see below). There was a target recruitment of 10 patients, and patients were approached in successive groups of 20 until the target was achieved.

### 2.2. The ACB Reduction Intervention

The primary care pharmacist undertook a refresher training session that provided information on the unwanted effects of ACB drugs, the ACoB scale and tools to support deprescribing, information on alternative drugs with lower ACB sores, and the study processes. This was delivered through video-recorded talks, which could be watched at a time convenient to the pharmacist. The pharmacist, who had full access to the medical record, including the diagnosis and medication history, reviewed the medication of each participating patient prior to agreeing on a time and date for a telephone consultation, during which the pharmacist discussed the need to deprescribe or switch a drug to reduce the ACB. The study pharmacist was an experienced, qualified independent prescriber. In line with the study protocol, they assessed the ACoB score, deprescribed and/or switched the drugs to an alternative, implemented the agreed change, updated the patient’s medical record, and completed a pharmaceutical care plan (PCP) (based on one used in a previous study; https://www.uea.ac.uk/groups-and-centres/chipps). The patient was advised about any new medications that they had been prescribed and informed about potential side effects.

No second consultation with the pharmacist or GP was routinely scheduled. Patients were asked to contact their pharmacist/clinician/GP if they had any subsequent changes in symptoms or were experiencing any unwanted side effects. The GP and the study team were then notified of any symptoms or medication changes by the pharmacist, who also updated the PCP. Patients were advised that in an emergency, they could contact the chief investigator (P.K.M.), a medical consultant in care of the elderly.

### 2.3. Data Collection and Management

*Response, recruitment, and retention*: The responsible clinician (or authorized depute) was asked to complete a standard form recording the total number of patients for whom they were responsible, the numbers identified as potentially eligible from the random sample of 200, the numbers excluded on screening and reasons, and the numbers invited with dates. Patients were identified by a unique identification (ID) number.

A.N. kept a record using a study log of the numbers invited, the numbers expressing interest, the numbers giving informed consent, the numbers attending the first consultation, the numbers providing baseline data, and the numbers providing follow-up data.

*Pharmaceutical Care Plans—Medication changes recommended, implemented, and sustained*: The PCP included details of the recommended changes and their implementation for each patient. PCPs were anonymized by the pharmacist prior to sending to the lead researcher (AN) at the end of the study period for data entry and analysis. Medication changes were categorized by drug groups (therapeutic class) and process measure (such as stop drug, change drug, reduce dose, etc.). The sustainability of the changes was assessed.

*Patient survey:* A questionnaire was developed and sent to the patients by post at baseline (immediately after the first consultation), 6 weeks, and 12 weeks (Appendix A). The baseline version included demographic data, consultation time, patients’ views on the intervention, and acceptability of intervention assessed by a Likert rating scale (5, strongly agree; 4, agree; 3, uncertain/not applicable; 2, disagree; and 1, strongly disagree). Quality of life (QOL) was measured by EQ-5D-5L (Appendix A). There wase also an open text response option in each section of the questionnaire for participants to add any further relevant information. At 6 and 12 weeks the questionnaires asked about their views of the intervention, any subsequent medication changes, their quality of life, and their views on the study processes (12 weeks only). Questionnaires were identified by ID number only. Patients were asked to complete and return the questionnaires to the CI, P.K.M., by pre-paid addressed envelope, who forwarded them to the lead researcher, A.N. Telephone calls were made to remind patients if the questionnaire was been returned within 2 weeks.

*Participant interviews*: Semi-structured interviews were conducted with a purposive sample of patients (different ages, sexes, and types of ACB medication changes) with a target sample size of five to fit within existing resources and the likely response rate. The primary care pharmacist and linked practice doctor were also invited to take part in an interview. A.N. contacted the interested participants to arrange a suitable date and time for interviews. The interviews were conducted virtually according to COVID-19 restrictions at the time, using phone or video calls.

An interview topic guide was developed specifically for each participant group (Appendix A). The interviews were digitally recorded and transcribed verbatim either by A.N. or a University of Aberdeen approved external transcription company (NJC Secretarial).

*Adverse Event and Adverse Reaction reporting procedure*: Patients were asked to self-report any new or unexpected symptoms to their pharmacist/GP who reviewed these and made changes to the prescribed medications and/or advised the patient where necessary. The pharmacist/GP then reported the event to the research team (and if initially reported to the pharmacist, to the GP) and the yellow card scheme (if necessary). A.N. and P.K.M. assessed each event for seriousness (adverse event, adverse reaction, serious adverse event, and serious adverse reaction), the likely causality, and whether it was expected or unexpected.

*Sample size, data management, and analysis*: Recognizing the high pharmacy staff workload during and after the COVID pandemic, the pharmacist was asked to recruit a minimum of ten patient participants. This was a pragmatic decision based on balancing pharmacist capacity with a sample sufficient to address the study objectives. Data from the questionnaire surveys were entered into an SPSS V.27 spreadsheet by A.N. Simple descriptive statistics were conducted for quantitative data. There was an initial plan to analyze open text responses thematically, but due to limited resources and few responses, these are not reported here. Interview data were transcribed and checked against the recording. Thematic analysis was used to generate themes and subthemes.

## 3. Results

### 3.1. Patient Identification, Recruitment, and Retention

From 200 patients screened, 41 potential participants were identified as potentially eligible. After further screening, nine patients were excluded due to severe mental illness or hearing difficulties. From the remaining 32, twenty patients were invited to take part in the study; four did not respond to the invitation, and two declined to participate (one had problems with understanding the study processes; one was due to have an operation during the study period). Fourteen patients gave formal consent and attended the consultation with a primary care pharmacist. One patient was excluded due to an unsuccessful previous attempt to switch medication. Thirteen baseline, 6 weeks, and 12 weeks follow-up questionnaires were sent to patients, and all of them were returned to the research team within two weeks. The number of patients in each step is presented in the CONSORT flow diagram (Figure 1). The response rate of participants was 16 of 20, the recruitment rate was 14 of 20, and the retention rate was 13 of 14.

### 3.2. Participant Demography, Outcomes, and Experiences of Intervention

Most patients were male (*n* = 8/13); the median (IQR) age was 72 years (68.5–75.5). The mean (SD) number of medications was 7.62 (3.18), and patients had an anticholinergic cognitive burden (ACoB) score of 3, 4, or 6 (Table 1). Patient views on the acceptability of the intervention as assessed by the Likert responses are presented (Table 2). All participants answered all questions, and all responses had face validity, confirming this data collection method was acceptable and feasible. All patients thought the duration of the consultation was appropriate, and they were happy with the pharmacist’s consultation and medication review. The ACB medication changes for individual patients are summarized (Table 3); most of medication changes involved a switch. At three months, one patient had reverted to their original medication, and two patients were on a combination of the new medicine and an occasional low dose of the original medication. The QOL as measured by EQ-5D-5L at baseline, 6 weeks, and 12 weeks is shown (Table 4). Overall, the scores showed few problems. The respondents reported the lowest scores for self-care and the highest in pain/discomfort, but the differences were small. The utility of EQ-5D-5L was calculated by the utility equation by taking the utility score of 1 (complete health) and subtracting the coefficients of the five dimensions. The values of the EQ-5D-5L utility [median (IQR)] were 0.592 (−0.515, 0.859), 0.667 (−0.402, −1.411), and 0.655 (−1.411, 1.000) at baseline, six weeks, and 12 weeks follow-up, respectively.

Most patients were happy to be randomized in a future study. Six interviews were conducted with five patients and one primary care pharmacist. One GP could not take part in an interview but provided written responses to the question by email. The themes and subthemes are shown (Table 5), illustrated by verbatim quotes identified by participant ID, sex, and age. Overall, the patients suggested the study processes were acceptable. They were happy about the medication changes and satisfied with the pharmacist consultation, and some reported reduced side effects compared with before the medication change.

Patients reported that peripheral adverse effects such as dry eyes, dry mouth, or constipation and central side effects such as sleepiness decreased after deprescribing or switching ACB drugs to the alternatives.

## 4. Discussion

The REGENERATE feasibility study showed that patients could be identified and recruited from a primary care setting by a primary care pharmacist. Seventy percent of the invited patient participants attended the pharmacist-led consultation, and all but one accepted the recommended changes and completed the baseline and all follow-up questionnaires. The data collection forms were suitable, and the data suggested that the intervention may be effective at reducing ACB. The participants’ survey responses and interviews demonstrated that the intervention was acceptable for both patients and healthcare professionals. Primary care may be an appropriate setting to conduct a future definitive trial, and participants would be willing to take part and be randomized.

Strengths of the study included the fact that it was based on the published literature and detailed qualitative work brought together in a logic model. The study showed a high response and retention rate, which might be due to good communication and a positive relationship between the primary care pharmacist and their patients, including the prior telephone conversation with the pharmacist before sending the invitation packs. Only one primary care pharmacist delivered the intervention in a single primary care setting, which was a strength as it could be assumed that the intervention was delivered consistently but also a limitation (see below). Fidelity to the intervention was supported by training and a detailed standard operating procedure, although without observation, fidelity cannot be assumed.

Limitations were that findings from one pharmacist at one site might not be representative of the wider primary care pharmacist population, and likewise, the small sample size limited the generalizability. Virtual conduct of the interviews using the telephone meant that non-verbal communication from patients could not be included. A potential solution to this is the use of other platforms such as MS TEAMS or Zoom meetings to conduct the patient interviews if face-to-face interviews are not possible. Further, the interview with the doctor had to be replaced by a free text survey, which provided less rich information than an interactive interview. Another source of bias in our study sample was the exclusion of patients with severe mental illness. This is a standard patient exclusion criterion for studies because of concerns including the research burden on vulnerable participants, issues of capacity, and the ability and/or motivation to comply with study processes. However, it is increasingly recognized that where possible, such patients should be given the opportunity to take part in studies. It was not considered necessary or ethical for them to be included in this early feasibility study of assessing the delivery of the intervention and not assessing the clinical outcomes associated with the service.

In our feasibility study the ACB score decreased from the baseline after a primary care pharmacist provided the intervention. In addition, during the interviews, patients reported decreased side effects of medication after receiving the intervention. Likewise, other studies have shown that a pharmacist, either individually or as part of a team undertaking patient medication review followed by recommendations to the prescriber, can reduce ACB [11]. However, in our study, one primary care pharmacist could implement medication deprescribing to patients successfully on their own due to being qualified as an independent prescriber pharmacist. Likewise, other studies have shown that the deprescribing can be done successfully by pharmacists [24,25,26]. Based on the utility calculation of EQ-5D-5L in all patients during the study period, small improvements in QOL were reported at six weeks’ follow-up, which may be due to patients having a positive attitude about the intervention or medication changes. However, improvement in QOL at 12 weeks follow-up decreased slightly compared with 6 weeks follow-up but was still higher than at the baseline. However, we reported these only as trends since a non-powered feasibility study cannot be used to assess effectiveness, and therefore, the changes were not tested for statistical significance.

Pharmacists may be well placed to provide this ACB intervention based on this study and as reported elsewhere [27]. The increasing use of pharmacists based in general practice and also in community pharmacy is supported in recent UK policy as way of managing workload in primary care and reducing pressure on GPs [28]. Deprescribing as a specific focus of medication review is also now well established [16], and the general barriers and facilitators are increasingly understood. However, deprescribing with the specific aim of reducing ACB is a highly specialized application that requires additional knowledge and judgment as most changes require a switch to an alternative medication. In 2026, all newly registered pharmacists in the UK will be qualified as independent prescribers, and the ability to implement changes themselves will facilitate deprescribing in general and ACB reduction specifically. However, it is important that pharmacists delivering the intervention are operating within their level of competence, and if necessary, any future trial must ensure the pharmacists are experienced in ACB or are provided with relevant training.

## 5. Conclusions

Deprescribing helps to promote the appropriate use of ACB medication. Pharmacists with independent prescribing rights are potentially able to deliver a successful deprescribing intervention, implementing recommended medication changes themselves with a potential to lower the anticholinergic cognitive burden score and increase patients’ QOL. This feasibility study has shown that patients can be recruited and retained and would be willing to take part in a randomized definitive trial. Data collection forms are suitable, and the results suggest that the intervention may be effective at reducing ACB and is acceptable to patients and health care professionals. The next stage of this program of work should be a randomized pilot study, followed by a definitive RCT to assess the effectiveness and efficiency of a pharmacist-led ACB deprescribing intervention compared with current practice.

## Figures and Tables

**Figure 1 geriatrics-09-00152-f001:**
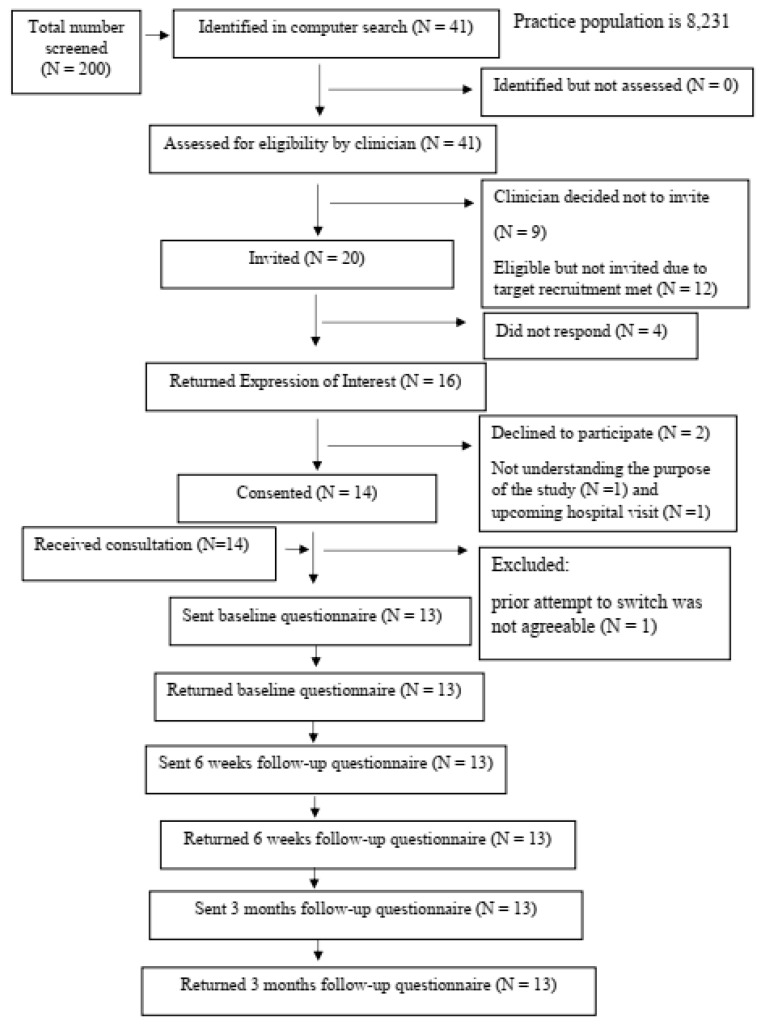
CONSORT flow diagram.

**Table 1 geriatrics-09-00152-t001:** Summary of patient characteristics at baseline N = 13.

Characteristics	N *
Males Females	8 5
Age in years 65–69 70–74 75–79	3 5 5
Number of medications (mean ± SD)	7.62 (±3.18)
Anticholinergic cognitive burden score 3 4 6	7 1 5
The median time (IQR) for the initial consultation with the primary care pharmacist	15 min (12.5–20 min).

* Due to small denominator (13), percentage is not reported.

**Table 2 geriatrics-09-00152-t002:** Patient participant responses to Likert * statements about their experiences and views at baseline, 6 weeks, and 12 weeks (*n* = 13).

Items	Baseline (Median; IQR)	6 Weeks Follow-Up (Median; IQR)	12 Weeks Follow-Up (Median; IQR)
The pharmacist appeared well informed.	5 (4, 5)	-	-
The pharmacist listened to what I had to say.	5 (4, 5)	-	-
The pharmacist answered all my concerns.	5 (4, 5)	-	-
I would rather have seen a doctor.	3 (2, 3)	-	-
I would rather have seen a nurse.	3 (2, 3)	-	-
Happy with my consultation with the pharmacist.	5 (4, 5)	-	-
Good idea to change medication to reduce chance of unwanted side effects.	5 (4.50, 5)	-	-
The study purpose was clear.	5 (4, 5)	-	-
Given enough information to decide whether to participate.	5 (4, 5)	-	-
**The new approach to your medication**			
Happy to discuss medicines with the pharmacist.	-	5 (4, 5)	5 (4, 5)
The symptoms of illness are controlled.	-	3 (2, 4)	4 (4, 5)
No concerns related to new approach for reviewing medicines.	-	4 (4, 5)	4 (4, 5)
Happy with the changes made to my medicines.	-	4 (3, 4.50)	3 (3, 5)
Unhappy with the changes made to medicines.	-	3 (2, 3)	3 (3, 3)
Currently on a new medication.	-	4 (3, 5)	4 (3, 4)
Changed back to my old medicines.	-	3 (2, 3)	2 (2, 3)
**About the study processes**			
The questionnaires were clear.	-	-	5 (4, 5)
The questionnaires were easy to complete.	-	-	5 (4, 5)
Happy to complete the questionnaires.	-	-	5 (4, 5)
Study participation did not take up too much time.	-	-	5 (4, 5)
Study participation did not give any stress.	-	-	5 (4, 5)
Consider participating in a future study examining similar issues.	-	-	5 (4, 5)
Recommend a friend to take part in this kind of study.	-	-	5 (3, 5)
Interested in being part of a patient advisory group for future studies.	-	-	3 (2, 5)
Further study should be encouraged in this area.	-	-	5 (4, 5)

* 5 = strongly agree; 4 = agree; 3 = unapplicable/uncertain; 2 = disagree; 1 = strongly disagree.

**Table 3 geriatrics-09-00152-t003:** Summary of individual patient ACB medication changes recorded in the pharmaceutical care plan at baseline, 6 weeks, and 12 weeks.

Participant ID	Sex	Age (Years)	ACoB Score Baseline	ACoB Score 6 Weeks	ACoB Score 12 Weeks	Old Treatment	New Treatment	Sustainability
P1	Male	72	3	0	0	Amitriptyline 25 mg tablets one at night	Pregabalin 50 mg capsules one three times daily	Patient remained on new medication
P2	Male	75	3	0	0	Hydroxyzine hydrochloride 10 mg tablets one at night	Fexofenadine 180 mg tablets one at night and added an emollient-Zerobase to see if helps itch	Patient remained on new medication
P3	Male	73	3	0	0	Tolterodine 2 mg capsules one daily	Mirabegron 25 mg tablets one daily	No improvement on 25 mg dose so increased to 50 mg and then remained on new medication
P4	Female	69	3	0	0	Solifenacin 5 mg tablets one daily	Mirabegron tablets 50 mg one daily	Patient remained on new medication
P5	Male	77	3	0	0	Chlorphenamine 4 mg tablets	Fexofenadine 180 mg tablets one daily	Patient remained on new medication
P7	Male	67	6	1	1	Cetirizine 10 mg tablets Amitriptyline 10 mg tablets	Fexofenadine 180 mg tablets one daily Pregabalin capsules 75 mg one twice daily	Patient remained on new medication but with pregabalin increased to 150 mg twice daily
P8	Female	70	4	1	1	Amitriptyline 10 mg tablets three at night Co-codamol 30/500 tablets	Reduce dosage to 20 mg over the next 2 weeks with a view to stopping altogether Naproxen 500 mg tablets one twice daily	The patient has stopped completely with no adverse effects The patient was managing with naproxen but due to struggling to sleep at night started taking occasional co-codamol
P9	Male	68	6	3	3	Chlorphenamine 4 mg tablets one three times daily Cetirizine 10 mg tablets once daily Amitriptyline 10 mg one at night	All previous medications stopped. Replaced by fexofenadine 180 mg tablets once daily	The patient was still not sleeping well. -chlorphenamine 4 mg tablets once at night added back Otherwise, patient remained on new medication with no adverse effects
P10	Male	76	6	1	4	Solifenacin 5 mg tablets Nefopam 30 mg tablets Co-codamol 30/500 mg tablets	Mirabegron 50 mg one daily Reduce dose of nefopam if able and replace with increasing dose of co-codamol, to a regular four times daily as opposed to when required	Mirabegron did not help urinary urgency at all. Stopped and reverted back to solifenacin but higher dose Patient stopped taking nefopam completely after increase in dosage of co-codamol
P11	Female	66	6	0	0	Hyoscine butylbromide 10 mg tablets Hydroxyzine 10 mg tablets	Mebeverine 135 mg tablets one three times daily Peppermint oil capsules 0.2 mL one three times daily Fexofenadine 180 mg tablets one daily	Patient remained on new medication
P12	Male	71	3	3	3	Amitriptyline 50 mg tablets	Amitriptyline 30 mg tablets Mirtazapine 15 mg tablets	Currently taking 35 mg amitriptyline as at lower dose sleep was disturbed Patient remained on new medication
P13	Female	75	6	0	0	Solifenacin 5 mg tablets Dicycloverine 10 mg tablets	Mirabegron 50 mg tablets Peppermint capsules/mebeverine 135 mg tablets	The patient remained on the new medication
P14	Male	79	3	0	0	Solifenacin 5 mg tablets two times daily	Mirabegron 50 mg tablets	The patient remained on new medication

Note: ACoB = anticholinergic cognitive burden.

**Table 4 geriatrics-09-00152-t004:** Patient participant EQ-5D-5L scores * at baseline, 6 weeks, and 12 weeks; *n* = 13.

Items	Baseline (Median; IQR)	6 Weeks Follow-Up (Median; IQR)	12 Weeks Follow-Up (Median; IQR)
Problems in mobility	2 (1, 3)	2 (1, 2.75)	1.50 (1, 3.25)
Problems in self-care	1 (1, 1)	1 (1, 1)	1 (1, 1.5)
Problems in usually activities	2 (2, 2.50)	1 (1, 2.75)	1 (1, 3.25)
Pain/discomfort	2 (1.50, 3.50)	2 (1, 3)	2 (1, 3.25)
Anxiety/depression	1 (1, 2)	2 (1, 2)	1 (1, 2)

* 1 = no; 2 = mild; 3 = moderate; 4 = severe; 5 = very severe.

**Table 5 geriatrics-09-00152-t005:** Summary of patient participant interviews: themes and sub-themes.

Themes	Subthemes/Exemplar Quotes
**Remembering the purpose of the study**	Patients remembered the purpose of the study, i.e., was involved in medication changes in people aged 65 year and over to new medication to reduce side effects and improve quality of life. *It was about trying to find out whether the medication that was routinely given to elderly people was still doing the best job*. [P8, female, 70y] *Well, it was about my medication and I got my medication changed, that’s what it was about*. [P4, female, 70y]
**Process of the study**	*What went well:* Many patients suggested the process of study went well. They were happy about the medication changes and satisfied with the pharmacist consultation. They kept in touch with the pharmacist when they needed further help or had problems about medication usage. *Well, it’s fine, I’m on new medication and it seems to be working out fine. The pharmacy has been in touch with me a couple of times and all’s well.* [P4, female, 69y] *Maybe … yes, just consultation or just touching base with the pharmacist, the pharmacist was good, and she did keep in touch but probably it would’ve been better to keep in touch just a little bit more, I think. But equally well I could have contacted her, she was available to be contacted so it wasn’t really a problem*. [P8, female, 70y] The pharmacist noted that patients were happy with the information pack that explained everything clearly and also additional telephone calls from the pharmacist. *The ones we picked were very positive about it, they were very interested. They appreciated the pack that you sent out with all that information which explained everything to them very clearly, although I had gone over very quickly the basics on the telephone. The fact that it was anonymous was good, they liked that idea*. [Primary care pharmacist] The pharmacist was happy to conduct the research in her role. Patients felt positive about pharmacists’ involvement in this type of study. GP also valued the pharmacist’s opinion as he was not required to provide further input toward the plan and no patients complain about the changes or anything else. *It was very good, I enjoyed doing it. The patients were very positive*,… [Primary care pharmacist] *Pharmacist team manage this, and I was not involved as they found no issues requiring GP input* [GP]
	*What did not go so well*： A few patients were excluded from the study because they were waiting for hospital admissions for procedures, and pharmacist felt that the timing was not right to make any changes due to their upcoming appointments. *One or two of them I think we decided weren’t eligible because of the other drugs they were on, or they were awaiting a procedure at the hospital, and we didn’t want to change anything before they went on for that. That was quite a valid criticism. It’s just circumstances really*. [Primary care pharmacist] The pharmacist suggested the monitoring paperwork was burdensome due to time consuming process and did not really fit to routine practice. She felt that six weeks is too long to monitor patients following medication changes. *I think part of your paperwork it said three months review, six weeks review; those didn’t really fit what I was doing. Six weeks was too long to leave them, I was phoning them maybe two weeks, three weeks, or they were phoning me and saying it wasn’t going well. A shorter timeline for the initial review would be better*. [Primary care pharmacist]
	*Difficulties in taking part in the study*: One patient thought the questionnaires were quite repetitive due to similar questions asked in each follow-up time. It could be that he misunderstood the quality-of-life monitoring in the long period. However, others suggested no difficulties in taking part in the study at all. *Actually, found the questionnaires that you sent were quite repetitive. You were asking the same information. That’s really about all, you know, I can say. I filled in several questionnaires, and I seemed to be answering the same questions*. [P2, male, 75y] *Not physically or anything. No, I was … as far as I can remember there were no difficulties in it at all*. [P7, male, 67y]
**Patient-related outcomes**	*General satisfaction and well being* In general, patients were satisfied with the pharmacist consultation and were happy with medication changes that reduced the side effects of the medication. *Well, it’s fine, I’m on new medication and it seems to be working out fine. The pharmacy has been in touch with me a couple of times and all’s well*. [P4, female, 69y] *When we first changed from my medication in the beginning, it was fine in that it helped me to sleep better, which was one of my big problems. After a while, the benefits seemed to wear off a bit and then my medication as changed again, which now suits me much better in all directions. It helps me to sleep better and it … yeah, yeah, the side effects, the constipation side effects if you like have gone. So that’s fine*. [P8, female, 70y]
	*Symptom control and side effects*: Overall, symptoms could be controlled better after medication changes. Patients were happy with a new medication or the alternatives that had fewer side effects. Patients reported that peripheral adverse effects such as dry eyes, dry mouth, or constipation and central side effects such as sleepiness decreased after deprescribing or switching ACB drugs to the alternatives *Well, yes, I kind of did wonder how’s this going to work? But you know this; I was happy to change to something else because I knew I wasn’t feeling great with the Buscopan. I know it does help for bloating and that, but I know when the lady said, “Take peppermint oil, I’ll try you with that”, I know peppermint is good for the stomach anyway. No, I was—the way I was feeling, Toney, I just wanted to try a change and see if it made a difference*. [P11, female, 66y] *Well, I’m not getting the same bloating the same. I do have IBS so I’ll always have that problem, but no, my tummy feels more comfortable from day to day. I’m quite happy to stick with that just now, yes*. [P11, female, 66y] *Yeah, for me it worked fine because my medication as changed. Now, I’m on something that suits me better with less sort of side effects… No, just … nothing worse, just better, just the symptoms were slightly better*. [P8, female, 70y]
**Suggestions for improving in the future study**	Patients valued regular review of medications and believed that it is quite important and needed. *Just a review, a regular review of the medication that you’re on and what it’s still doing rather than just assuming that everything is fine, and it continues to be fine*. [P8, female, 70y] The pharmacists suggested that the recruitment process could be improved in the definitive study. *I suppose that cut out a certain amount of the population which is a shame in a way because some of the other people we might have been able to help more, but it would’ve meant going through their family, and I felt because it was a pilot, that perhaps that was a step too far on this occasion. Perhaps when you’re doing the full study you could consider people like that, but we would need to involve families and that just makes it more complicated*. [Primary care pharmacist] Pharmacists also suggested paperwork needed to be improved to fit routine practice in the future direction. In addition, the initial monitoring period may need to be shortened to 2 weeks for close follow-up. The information pack was felt to be appropriate for use in the definitive trial. *I think part of your paperwork it said three months review, six weeks review; those didn’t really fit what I was doing. Six weeks was too long to leave them, I was phoning them maybe two weeks, three weeks, or they were phoning me and saying it wasn’t going well. A shorter timeline for the initial review would be better*. [Primary care pharmacist] *Well, I think your paperwork could be slightly better, I found it a bit confusing you know, where to put things. I mean other than that, not really, no because I think your explanation package was very good, the education stuff that you sent me, the examples of what to change, you could have more of that, you could expand that list I think, to help people*. [Primary care pharmacist] The pharmacists also suggested that well-qualified and experienced pharmacists may be best suited for conducting the definitive trial or when its rolled out into practice to avoid confusions and complications related to authorizations and mistakes. The trust of GPs for experienced pharmacists is quite important as the study did not need input from GPs. *I think that to some extent depends on how experienced the pharmacist is and where their place in the team is. If you were a newly qualified pharmacist, you’d just arrived, you didn’t know anybody, you didn’t know the doctors, I think it would be more difficult. I would imagine if you’re going to roll this out, say somebody is going to do research like this, you would want to go for pharmacists who are well established and almost certainly have to be prescribers so that they can just get going, otherwise you’ve got that interface between the pharmacist, the patient and the doctor or the nurse and that just complicates the whole picture quite honestly*. [Primary care pharmacist]
**Willing to take part in a future trial**	All patients, a primary care pharmacist, and a GP expressed their interest in taking part in future definitive trials.
**Experiences in a training session for a pharmacist**	The pharmacist expressed the importance and value of the training session, and its contents were quite important and extremely useful for her. In particular, it provided a specific example of an alternative, which made the decision-making process easier. *Extremely useful. I think I would’ve struggled with the time constraints that I had to actually achieve as much as we did. It was very useful to have specific examples of what one drug could be changed to and the reason for that, it just made it easier for me just to go ahead and do things as opposed to having to think about it. If I had to work it all out for myself, it would’ve taken me longer and again, we’re back to time*. [Primary care pharmacist] *Yes, yes, because the training gave you specific examples of what you could do, so we tended to search on the drugs that we knew we had a very simple, straightforward alternative to. That made the process, the decision-making process easier, I think*. [Primary care pharmacist]

## Data Availability

The data presented in this study are available on request from the corresponding author due to ethical reasons.

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
