# Peer review of "The REGENERATE Study: A Non-Randomized Feasibility Study of an Intervention to REduce anticholinerGic burdEN in oldER pATiEnts"

_geriatrics, 2024, doi:10.3390/geriatrics9060152_

Round 1

Reviewer 1 Report

Comments and Suggestions for Authors

Overall, an interesting read and the authors should be congratulated for their work. This is a well-conducted feasibility study that addresses an important issue. However, improvements in the justification of the study design, sample size, and statistical analysis, would benefit the paper.

Some specific comments follow:
1. The CONSORT flow diagram image resolution can be improved Introduction: Although it is acknowledged that "no definitive trials" have examined ACB-focused deprescribing, a broader literature review on existing deprescribing studies, even those not specifically targeting ACB, would provide more context.

Material and methods: in my opinion, there should not be primary and secondary objectives; a trial must have clear objectives, all valuable in the same manner; If authors do not state main objective and specific objectives, then all objectives should be treated as equals;

Methods: Although the sample size was pragmatically chosen, the text should clarify if the sample size was based on calculations and their assumptions. Please discuss how the sample size was determined and the potential for underpowering the study.

Methods: Authors should discuss why no statistical significance testing was performed on the outcome measures, even if it is a feasibility study! Some form of basic inferential analysis (e.g., t-tests or non-parametric alternatives) could provide preliminary insights into the intervention's effectiveness.

Author Response

Response to comments made by Reviewer #1 was prepared in a table, please find attached.

Reviewer 2 Report

Comments and Suggestions for Authors

This is a non-randomised feasibility study of an intervention to reduce anticholinergic burden in older patients to inform a subsequent definitive trial. The results demonstrated that a deprescribing anticholinergic burden intervention in older people is feasible in a UK primary care setting and may benefit patients.

General concept comments

The primary objectives of this study were to test processes of patient participant identification, to assess recruitment and retention rates of patient participants, and to review potential outcome measures and data collection processes. The response rate of participants was 16 of 20 (80%), recruitment rate was 14 of 20 (70%) and retention rate was 13 of 14 (93%). The secondary objective was to explore the acceptability of the intervention to patients and healthcare professionals. All patients thought the duration of the consultation was appropriate, and they were happy with the pharmacist’s consultation and medication review. Overall patients suggested the study process were acceptable. They were happy about the medication changes, satisfied with the pharmacist consultation, and some reported reduced side effects compared before the medication change.

The objectives were achieved. However, to be published, some issues should be addressed.

Specific comments

1.   Introduction

1.1. Anticholinergic burden is defined as the cumulative effect of using one or more medicines with anticholinergic effects. However, drugs with anticholinergic effects refer to both drugs that bind exclusively to muscarinic receptors (e.g., oxybutynin, trihexyphenidyl, and ipratropium bromide) and drugs whose anticholinergic activity is not connected with their primary therapeutic purpose and mechanism of action (e.g., antidepressants, antipsychotics, and antihistamines). In lines 50-52 the term ‘anticholinergic medicines’ should be clarified in relation to what it refers to. Furthermore, deprescribing only makes sense if it is clear which medicines should be withdrawn or their dose reduced (lines 57-63). For instance, see Lavrador, M. et al. A Universal Pharmacological-Based List of Drugs with Anticholinergic Activity. Pharmaceutics 2023, 15, 230.

1.2. For better understanding of the purpose of this feasibility study, the remaining areas of uncertainty that need to be resolved (lines 69-70) must be stated in the Introduction. For instance, in the ACB intervention logic model presented in Supplemental material 1, it is mentioned that there is no consensus who is the best person to deprescribe ACB drugs, but nothing is said in the Introduction section about this.

2.     Material and Methods

2.1. Line 87 – Different systematic reviews have identified many anticholinergic burden scales and indexes. They all differ in their origin, content, and how they quantify the anticholinergic activity of included drugs. Why did the authors choose the Anticholinergic Cognitive Burden (ACoB) scale? The last update of ACoB Scale was published in 2012 and includes 99 drugs (N. Campbell, I. Maidment, C. Fox, B. Khan, M. Boustani, The 2012 update to the anticholinergic cognitive burden scale, J. Am. Geriatr. Soc. 61 (2013) S142–S143) but the authors use the 2008 ACoB Scale (ref. 18). Why?

2.2. Lines 88-92 – The exclusion criteria included patients with severe mental illness, but ACoB Scale tries to identify exactly the severity of anticholinergic effects on cognition of drugs, like their association with cognitive function in older adults, specifically delirium, mild cognitive impairment, dementia or cognitive decline. The authors should comment this potential bias.

2.3. Line 108 – What’s the meaning of ‘training on ACB’? This methodology must be presented. Were only drugs on the ACoB scale considered?

2.4. Lines 108 and 112-113 – It is not clear if the primary care pharmacist and the qualified Independent Prescriber pharmacist are the same professional or two distinct professionals. I know that lines 271-273 refer that there is only one pharmacist, but it should be better explained in the Methods section.

2.5. Line 112 – Does the pharmacist proposing a change in therapy have access to the patient's active diagnoses or just their pharmacotherapeutic profile? For instance, amitriptyline can be replaced by sertraline in depression but, in the case of neuropathic pain, it would be more appropriate to replace it with pregabalin. Without knowing the diagnosis, it is not possible to know the best option to take. It must be clarified in the paper.

2.6. Line 115 – This site cannot be found: https://sites.uea.ac.uk/chipps

3.     Results

3.1. Line 183 – Of the remaining 32 potential participants, 20 were invited to participate. How was this selection made?

3.2. Line 236 – The title of table 5 must appear separate from the caption of the previous table.

4.     Discussion

4.1. Lines 267-268 – «Patients reported decreased side effects of medication after receiving the intervention». Were only anticholinergic drugs deprescribed when adverse effects were reported or were these drugs deprescribed regardless of whether adverse effects were present or not? The results of side-effects reported at baseline and at the end of the study must be presented in the Results section to confirm this decrease. What types of adverse effects are the authors referring to? The ACoB Scale tries to identify the severity of anticholinergic effects on cognition of drugs (delirium, mild cognitive impairment, dementia or cognitive decline). And what was assessed in terms of peripheral anticholinergic adverse effects?

Author Response

Response to comments made by Reviewer #2 was prepared in a table, please find attached.

Round 2

Reviewer 2 Report

Comments and Suggestions for Authors

The authors' responses to my comments were satisfactory. The only thing left to explain was the use of the 2008 version of the ACoB scale instead of the 2012 update.